# Woodland Expansion in Upland National Parks: An Analysis of Stakeholder Views and Understanding in the Dartmoor National Park, UK

**Olivia FitzGerald** * , **Catherine Matilda Collins** and **Clive Potter**

The Centre for Environmental Policy, Imperial College London, The Weeks Building 16-18 Princes Gardens, London SW7 1NE, UK; t.collins@imperial.ac.uk (C.M.C.); c.potter@imperial.ac.uk (C.P.)
* Correspondence: oliviafitzgerald@hotmail.co.uk

**Abstract:** Woodland expansion on a significant scale is widely seen to be critical if governments are to achieve their net zero greenhouse gas ambitions. The United Kingdom government is committed to expanding tree cover from 13% to at least 17% in order to achieve net zero by 2050. With much lowland area under agricultural production, woodland expansion may be directed to upland areas, many of which are national parks under some degree of conservation jurisdiction. This may prove to be controversial, requiring full engagement with the interests of those individuals with a stake in their protection and management. In this paper, we explore how a range of stakeholders view the prospect of woodland expansion in Dartmoor National Park in southwest England, UK. Fifteen stakeholders—a mix of key informants and farmers—were shown different woodland expansion scenarios in map form and consulted using semi-structured interviews. The findings suggest widespread enthusiasm for woodland expansion, but with significant differences in terms of the scale and approach. Stakeholders raised topics of biodiversity gain, climate change mitigation, environmental benefits, cultural ecosystem gain, and forest crop benefits. Caution was expressed regarding target setting, the place of woodland expansion in the national debate, and the potential for harm from inappropriate new planting. The constraints identified were land tenure patterns, notably tenancy insecurity and 'common land' challenges, historical farming policy and culture, landscape objectives, and future policy design.

**Keywords:** climate change; carbon sequestration; biodiversity; stakeholder engagement; ecosystem services; tree planting

## 1. Introduction

Committing to achieving net zero emissions is a widely shared ambition amongst those countries that were signed up to the Paris Agreement on Climate Change [1]. In 2019, the UK government legislated to bring all greenhouse emissions to net zero by 2050 and the Climate Change Committee (CCC), the UK's independent climate advisory board, has outlined a net-zero strategy [2,3]. As in other countries, this strategy involves both emissions reduction and the sequestration of any remaining emissions [3]. A significant element of the sequestration strategy involves an increase in UK tree cover from 13% to at least 17%, which equates to planting at least 30,000 hectares per year from 2024 until 2050. Two further reports exceed these tree cover recommendations: the Centre for Alternative Technology's (CAT) 'Zero-Carbon Britain' report recommends 24% tree cover and Friends of the Earth (FOE) suggest doubling UK tree cover to 26% [4,5]. In the UK, 77% of land use is agricultural [6] (p. 23), and the influential National Farmers' Union has also produced a net-zero report, signaling a move towards tree planting [7]. The NFU proposes that 'increasing farmland woodland could deliver GHG savings of 0.7 MtCO2e/year', and it indicates that the farming community is open to increasing tree cover, but without specific targets [7] (p. 8).

However, large scale woodland expansion is not without its critics, and some commentators have called for caution in the global rush to plant trees [8–11], suggesting that an over-reliance on the ability of trees to sequester carbon may distract from efforts to reduce emissions, and that planting in some areas may lead to inappropriate land use change and an export of emissions and deforestation [12,13]. Others argue that the ability of trees to lock up carbon may have been over-estimated, and that effective carbon lockdown will only be achieved as part of a crop forestry system that includes planning for the sustainable and long-term use of timber products [14–16]. Conservationists warn against narrowly focusing on carbon when using nature-based solutions, emphasising a holistic approach that avoids disrupting natural processes or damaging protected habitats [11]. Other concerns include reference to the skills gap in woodland management for keeping trees in productive health, the time that is taken for new trees to sequester carbon, the reduced effectiveness of ageing trees, and the effect that warmer temperatures and pests will have on forestry [6,14,16].

The issue of how 'best' to increase tree cover is not new. With many lowland areas in active agricultural production, upland areas are likely to be targeted. Upland area afforestation in the mid-twentieth century encountered resistance and caused substantial ecological damage in places, such as Scotland's Flow Country, with the UK Forestry Commission (FC) being widely condemned for the resulting 'Sitka slums' [17–19]. Yet, the CCC has called for commercial forests to expand production from their current output of eight million oven-dried tonnes (M odt) of products to 18–29 M odt by 2050 and for more timber to be used in construction 'to displace emissions' [6] (p. 43). Where this increased timber production should be located, and what type of wood should be produced, are now critical decisions. The expansion of woodland in national parks and other protected areas may prove to be particularly controversial. Traditionally seen as open landscapes that are centred on livestock farming, the effect of large scale planting is likely to be a shift in the structure and appearance of locations, often in the uplands, which are cherished for the close association between extensive farming systems and open habitats [20–22]. In addition to conservation, UK national parks have responsibilities for preserving cultural heritage and providing a leisure space and landscapes for people to enjoy [23]. Even though some afforestation in many of these areas could be seen as part of a broader restoration of former landscapes, careful consideration of their present ecology is essential for avoiding harm [11,22,24]. A recent review of uplands coverage identified a lack of research on the impact that increased tree cover could have on multiple ecosystem services, including cultural services, which these landscapes provide [22]. Burton et al., meanwhile, call for research to explore the trade-offs between ecosystem services and competing landscape objectives in specific contexts to inform afforestation policy at landscape scale [25]. Bonn et al. have suggested that current landscape preferences and habitat composition may need to be challenged [26], and it has been speculated that 'landscape tastes will prove dynamic' as people begin to favour features that contribute to carbon-neutrality [27] (p. 158).

There is a particular knowledge gap surrounding the social and cultural impacts of increasing woodland, with few studies on negative impacts, such as lower land values, reduced agricultural land area, and altered biodiversity [22,25,28]. Reviewing his re-wilded vision of the British uplands, Tokarski and Gammon claim that George Monbiot 'provides an entry point into a conflict between environmentalism and heritage that is not being debated adequately' [29] (p. 152). This debate might take different forms, depending on which UK nation (England, Northern Ireland, Scotland or Wales) is involved. Forestry is a devolved matter, with each nation being able to set its own policies amid its particular land tenure and landscape planning contexts.

How national park authorities will respond to a potential tree cover increase is unclear, although change in land use seems likely to be controversial given traditional views regarding open landscapes and the role of farming in sustaining them. In the UK, farming is the major land use in almost all national parks and it represents much of the heritage of these places. Exploring and understanding the views of a range of immediate and

local stakeholders and explicitly including farmers in meaningful dialogue is essential in determining the environmental and cultural future of national parks.

National parks certainly have potential for tree cover increase [30], although a lack of research on social and cultural impacts of woodland expansion is concerning [25]. This paper seeks to address this gap in the debate by mapping, identifying, and exploring the views of stakeholders whose future engagement is likely to be fundamental to progress. We use semi-structured interviews with key informants and farmers to explore their responses to how the CCC's tree cover recommendations might be applied in a national park setting. Dartmoor, in the southwest UK, was selected as a case study, although it is hoped that this method may be adopted elsewhere and that the findings will inform debate on an issue demanding urgent attention to mitigate further climate and ecological damage.

## 2. Materials and Methods

A literature review highlighted the key issues that are associated with woodland expansion in national parks; spatial GIS modelling allowed afforestation scenarios to be mapped; and, a case study approach enabled an in-depth study examining how people and processes interconnect [31]. Ethical approval was gained from the Science, Engineering and Technology Research Ethics Committee (SETREC) of Imperial College London.

### 2.1. Case Study Site

Dartmoor is a largely granite-based upland area in the southwest of England (50°34′ N, 4°0′ W). With a population of 34,500 people, approximately 90% of its land is agricultural and it is a popular tourist destination [32]. The case site was suitable as the 12.5% tree cover is close to the UK national park median (16.4%) and the Dartmoor National Park Authority's (DNPA) draft Management Plan was open to public consultation at the time of study. This contained an explicit question asking whether there should be a woodland cover target; thus, DNPA members and stakeholders were currently engaged with the topic. Of the 954km$^2$ that comprise Dartmoor national park, 37% is common land. In accordance with common land in England and Wales, this is privately owned, but other locals have certain common rights, such as livestock grazing and turf cutting for domestic use; there is also provision for some public access on foot and on horseback [33].

### 2.2. Afforestation Scenarios

Scenarios representing three different levels of tree cover were illustrated in map form using ArcGIS (Figure 1) [34]. Visualisation is recommended for future scenario research [35]—something that maps can partially achieve. The maps depicted the current tree cover (12.5%), 'Business As Usual' (BAU), the CCC's minimum recommendation of 17%, and CAT's 24% recommendation. Two approaches for achieving these were represented to stimulate conversation around expansion pattern; these were (a) focusing on planting trees along river catchments and (b) focusing on block plantings.

### 2.3. Stakeholder Selection

A stakeholder matrix (Figure 2) depicting levels of interest and influence was compiled in order to ensure that relevant and representative parties were included. In an iterative process, with feedback and further selection happening simultaneously, this was shared with three stakeholders for comment [35].

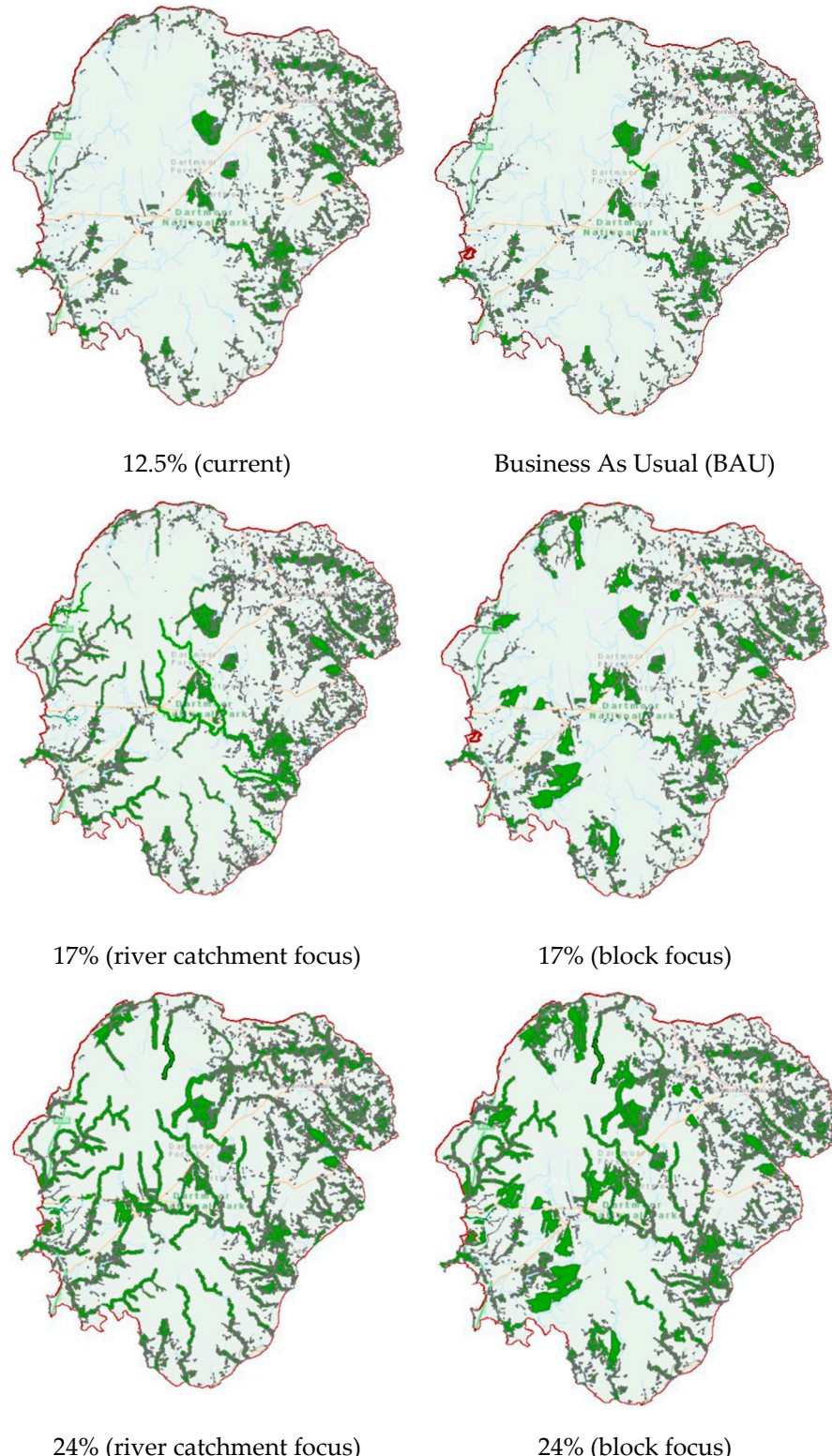

12.5% (current)

Business As Usual (BAU)

17% (river catchment focus)

17% (block focus)

24% (river catchment focus)

24% (block focus)

**Figure 1.** Tree cover maps of the Dartmoor National Park, UK, used to stimulate scenario-related discussion with the stakeholder interviewees. Two patterns of expansion, along river catchments and in contiguous blocks, as well as two target covers were used.

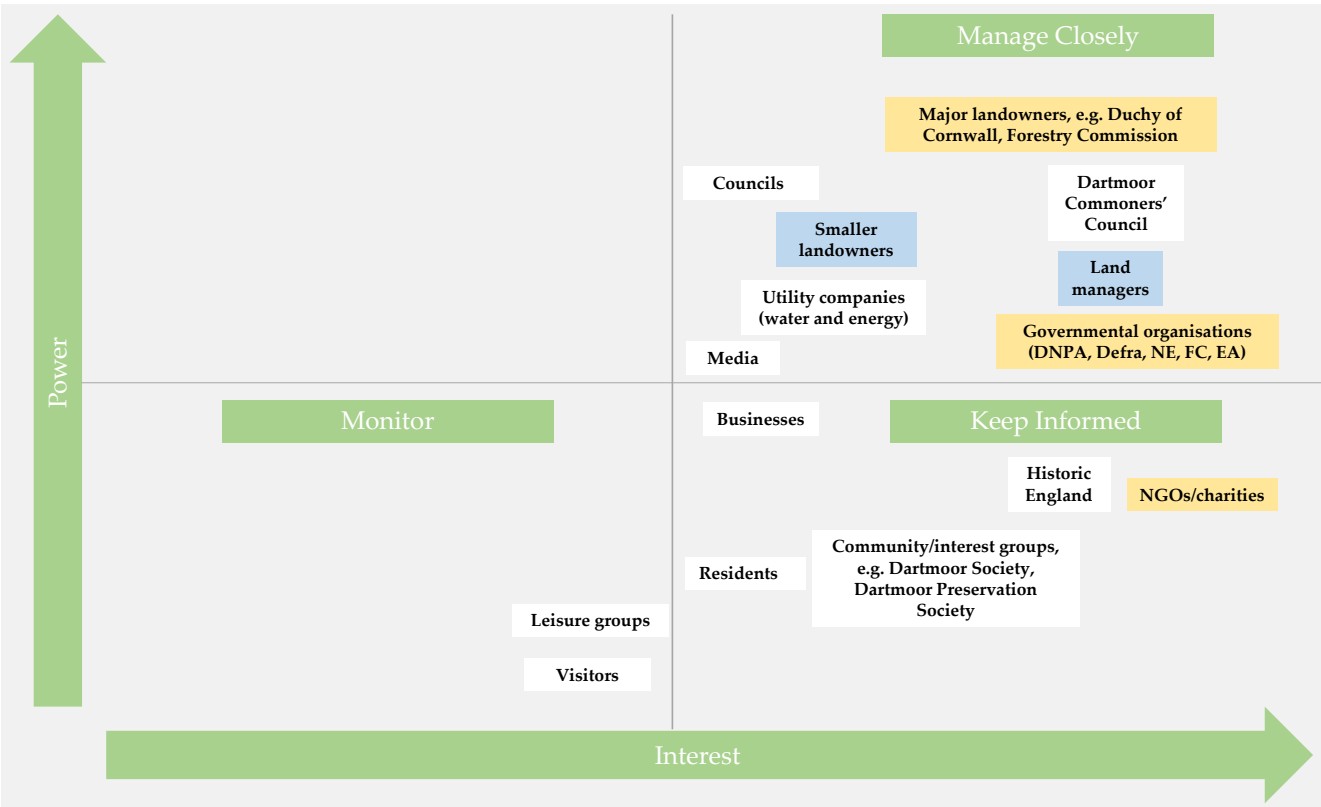

**Figure 2.** A stakeholder matrix based on axes of influence and interest mapped the stakeholders of the Dartmoor National Park, UK. Interviewees are representative of the blue (farmers) and yellow (key informants) stakeholder boxes.

A range of participants was explicitly recruited for the study in order to mitigate the bias inherent in the use of electronic approaches (i.e., that email responders are likely to be self-selecting as a group) [36]. Nevertheless, the data reflect the views of the group that was brought together for this work, not necessarily those of all Dartmoor farmers and interested players. A sympathetic manner was adopted to encourage participants to share their views and, although a neutral approach was attempted, it is impossible to entirely remove subjectivity from thematic analysis [37]. A close adherence to Braun and Clarke's [38] thematic analysis principles is hoped to have reduced this.

### 2.4. Interviews

Fifteen stakeholders were recruited: eight key informants (KI) (interested parties from the matrix population) and seven farmers (FA), and a process of semi-structured interview was used due to the scope for flexibility whilst maintaining a core focus [35]. The participants were remotely interviewed (between 29 June and 22 July 2020), interviews were recorded, transcribed by Otter.ai software, and then summarised for further analysis. All of the participants were sent a summary of their responses and key quotes to confirm accuracy and validity of the data and any interpretation [31].

### 2.5. Analysis

Thematic analysis was used to reveal recurring themes and patterns in the interview data [38]. A theoretical approach was adopted to focus attention on data related to the research question, thus only data that were relevant to the research question were coded. Manual coding ensured that non-verbal cues were included. The themes in this analysis comprise data that are relevant to the research question, prevalent through either frequency or via significance attributed by the participant.

## 3. Results and Discussion

The key informants engaged more readily with the process than farmers, for whom the issue of increasing trees may be a less immediate concern than the daily running of an agricultural business. However, there was wide engagement within the interview format, and the exercise generated a range of views and perspectives.

### 3.1. Section 1: Views on Woodland Expansion

There was near consensus that Dartmoor's tree cover should be expanded, with just one farmer indicating that tree cover should stay the same. The strength of feeling that was expressed by those interested in tree cover expansion ranged from almost-reluctant acceptance to lively enthusiasm. Thematic analysis led to the identification of three main themes: enthusiasm for tree cover expansion, caution over how this is achieved, and a willingness to increase tree cover that has been constrained by structural factors.

#### 3.1.1. Theme 1: Enthusiasm

Out of the 14 interviewees desiring tree cover expansion, eight demonstrated enthusiasm for dramatic expansion: five participants (3KI, 2FA) selected the 24% tree cover scenarios, two participants (1KI and 1FA) opted for even greater ambition, and one KI shied away from choosing or specifying a figure, but expressed that large-scale change is needed. Only farmers in the group favoured BAU or current levels (Figure 3).

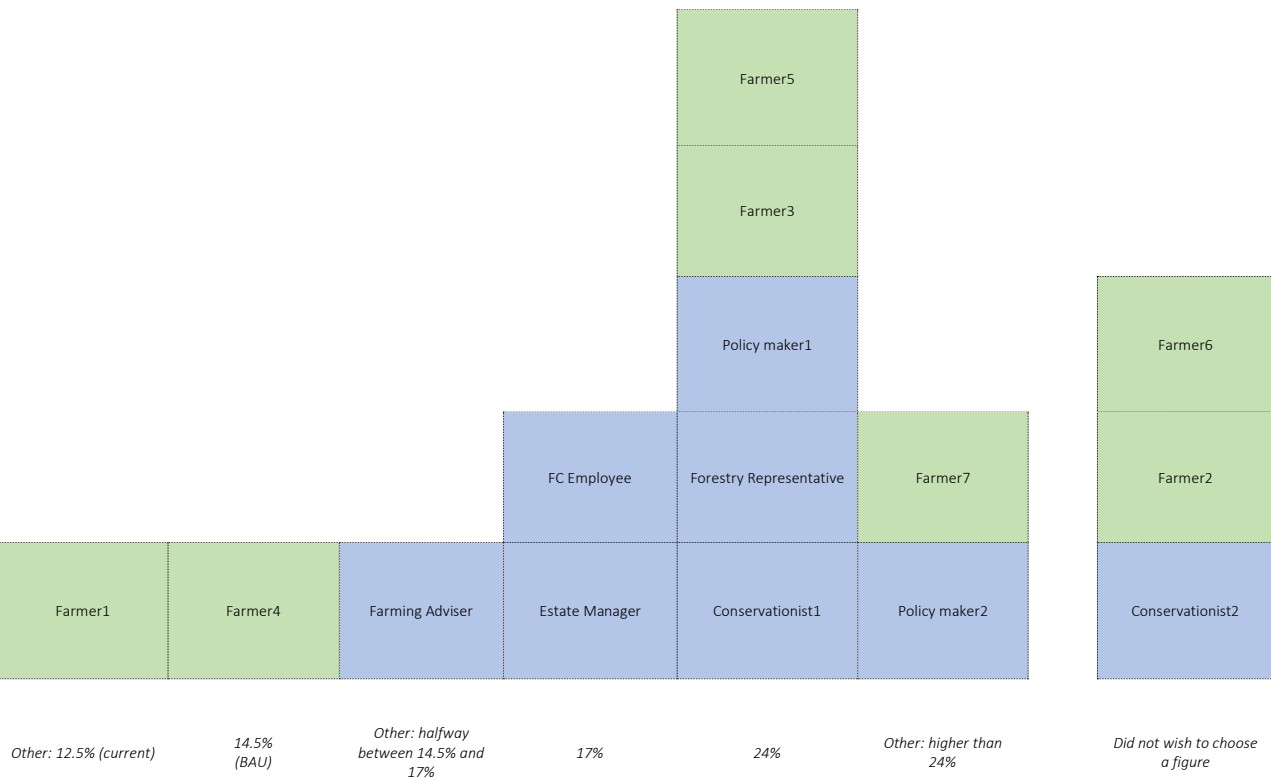

**Figure 3.** Afforestation scenario selection of the 15 stakeholder participants in semi-structured interview. In blue are key informants and in green are farmers.

Biodiversity

Enthusiasm for increasing tree cover was motivated by both a recognition of the many benefits that it provides, particularly from key informants, and a sense that a current lack of tree cover needs to be remedied. Concern at the state of nature within the national park was expressed by key informants with conservation backgrounds and by a farmer, who, talking about the current tree cover, demonstrated strong environmental motivations:

*"To say it's [increasing tree cover] a no-brainer is sort of under-egging it . . . I think it's a disgrace really".*

All other participants, except for one farmer, cited the need to improve biodiversity. This strong consensus reinforces the national park statutory duty of conserving wildlife [23] and the guiding principle of prioritising nature in the case of conflict [39]. Despite the potential ecological risks, both of the groups demonstrated a clear consensus that woodland expansion could positively contribute to local ecology.

### Climate Mitigation

The role of trees in mitigating the climate crisis was mentioned by every key informant, with most mentioning 'carbon sequestration' or 'climate resilience'. While most of the farmers discussed climate, only two mentioned 'carbon sequestration' or the 'climate emergency'. Despite the stresses that climate change is predicted to place on farming [40], the seemingly remote nature of climate change might explain why farmers discussed the issue less extensively than the key informants, many of whom have a professional responsibility to act upon the climate threat. Reassuringly for those warning against focusing narrowly on the carbon agenda [8–12], no participant spoke exclusively about carbon sequestration. From these interviews, it seems that carbon-centricity will be avoided by stakeholders' own reticence regarding such a narrow approach, along with their recognition of the wider benefits of woodland expansion.

### Environmental Benefits

In addition to carbon sequestration, other environmental benefits, such as flood mitigation and improved water quality, were most frequently cited by key informants. The broad environmental benefits of increasing cover were extensively covered by farmers who showed a high regard for these. Overall, stakeholder enthusiasm seemed to be most prominently motivated by the environmental and ecological benefits that trees provide. This echoes the many voices calling for tree planting to take its place in addressing both the climate and ecological crises [21,41,42].

### Farming Benefits

The potential farming benefits of woodland expansion did not appear as a key motivation for farmers, which suggested either that trees are seen to generate limited benefits for farming on Dartmoor or that farmers have not yet considered how best to take advantage of the woodland and trees on their land. This could also be symptomatic of historical agricultural policies that has tended to discourage trees from the farming landscape. 'Shelter' was the most frequent farming benefit mentioned:

*"We lamb all of our sheep outside, so we do appreciate . . . [our] very big set of conifer trees in one of the fields . . . .it's bloody brilliant when the wind comes in and you're outdoor lambing because the sheep get right under there—then it helps a lot".*

The key informants, five of whom have an agricultural role, spoke more enthusiastically about the farming benefits of trees, which suggested that the benefits of wood pasture are not always understood by farmers themselves.

### Cultural Ecosystem Services

Cultural ecosystem services, such as aesthetics and recreation, were recognised by both groups. Key informants frequently cited the ability of trees to absorb visitor numbers and ease pressure on popular sites. One farmer, who also runs a B&B, described how she, *"send[s] [guests] down to walk through the woods if it's raining".* This gives a brief insight into the tourism benefits offered by trees—a topic that is beyond the scope of this study but undoubtedly relevant to the issue of woodland expansion. The appeal of trees to visitors was echoed by a key informant who receives visitors throughout the year:

*"I thought about what [24% tree cover] would look like and I thought, 'my god, that'd be fantastic!'"*

Forestry Benefits

The groups differed most in their assessment of the forestry benefits that are offered by trees. Five key informants discussed benefit, such as increasing timber self-sufficiency, biomass and additional sources of income for land managers, whereas only two farmers mentioned these. The need to expand the UK's timber industry is detailed by the CCC's Land Use report [6], but the report's Chair highlights the contrasting skillsets of farmers and foresters [16]. Unless farmers are interested in acquiring forestry skills, or sub-contracting, they may not perceive forest-crop benefits of increased tree cover.

3.1.2. Theme 2: Caution

Concern over the global rush to plant trees was repeated in nearly all interviews [8–12]. This caution revolved around three things: target aversion, concern over the current national debate, and the potential for harm (Figure 4).

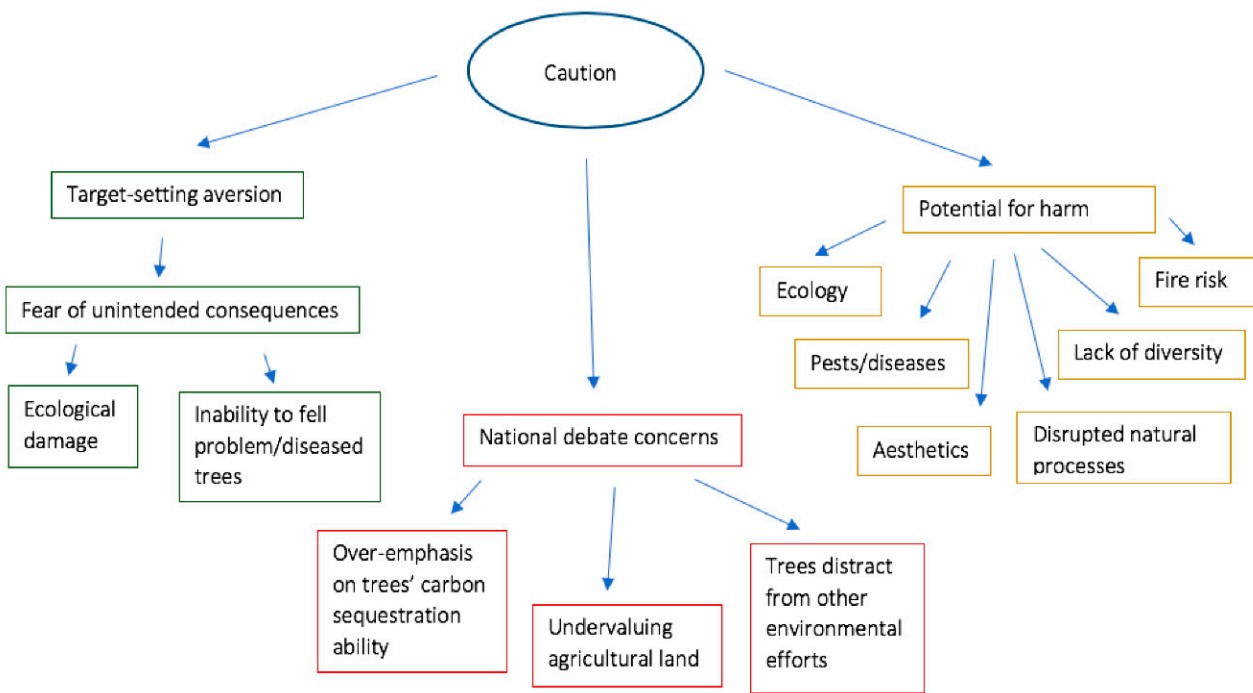

**Figure 4.** Thematic map of the cautions expressed by farmers and key informants discussing increasing tree cover in the Dartmoor National Park, UK.

Target Aversion

When choosing a tree cover scenario, three interviewees (1KI, 2FA) refrained from identifying any specific percentage, and three (2KI, 1FA) made a selection, but expressed concerns regarding target-setting. The aversion to targets was justified by worries about the unintended consequences of a large scale, target-driven expansion of tree cover, such as causing ecological harm or the inability to fell problematic trees:

*"I think targets are useful as discussion points, but they just open up a whole load of questions". (KI)*

*"I think if you're trying to get to a particular percentage, you make mistakes". (FA)*

*"That might then mean you can't take down, you know, some tree here". (FA)*

There was a sense from one key informant that targets can be useful on a national scale, but are more problematic at the regional or local scales. Conversely, some participants

actively supported target-setting, seeing it as a useful way of communicating aims and instigating action.

> "I think we do need some form of targeting or prioritisation". (KI)

> "I think we need to have a long-term target because, I mean, I've always found that in business, unless you set yourself a demanding target, you're never going to rise to the challenge". (KI)

Target-aversion appeared not to be motivated by a desire to halt progress, but rather to avoid potential harm. Two participants clearly expressed the utility of targets, thus, if one is selected, then clear parameters and conditions should be imposed based on prior experience to ensure that the risk of unintended consequences is reduced.

National Priorities Concern

Five interviewees voiced concern, raised in the literature, that there may be an over-reliance on tree-planting within climate policy (4FA, 1KI) [12]. Also mentioned were *"unrealistic expectations of what trees can achieve"* (FA), undervaluation of agricultural land (1FA, 1KI), and both corporations and farmers using trees as *"an easy environmental win"* (FA) to justify sub-optimal environmental behaviour elsewhere. It is clear that some feel that disproportionate attention is focused on the environmental impact of agriculture, rather than its productivity [43,44]. Concern, as outlined by Brown [13], that a rush to expand tree cover could lead to inappropriate land use change was raised by two farmers, highlighting the potential impact tree expansion could have on food production, with environmental problems being exported elsewhere:

> "If you double the tree cover of Britain, it would drastically reduce our ability to produce food". (FA)

> "It depends how we go with food production—whether we're bringing it in from abroad and blaming everyone else on the environment problems or keeping it here". (FA)

A key informant anticipated this issue, sharing Macdonald's view that food production should not be a main driver for places, like Dartmoor [42]:

> "If you're going to look at ten different environments for producing food, Dartmoor wouldn't be in those ten".

There is evident conflict here, with the farming and conservation sectors wishing to protect their interests. The single-issue focus on trees and tree cover may be unhelpful in this sense, with respondents expressing concern that their interests may not be fully integrated into proposals. The CCC's Land Use report [6] proposes an integrated approach towards achieving net zero, and this was articulated by one farmer, who commented "there's lots of pieces to this puzzle to get this net zero". Translating the national planning of the CCC report to regional scale will enable land managers to see how they fit in. A publicly agreed map indicating areas for woodland expansion, as proposed by five participants, would be an effective way of communicating this.

Potential for Harm

When asked why they thought that it was important to discuss tree cover on Dartmoor, many participants said that they thought tree cover expansion should happen, but that it should be done with care. This likely reflects collective memories of the negative effects of large scale tree planting in the past, notably in Scotland's Flow Country and, more recently, in the Lake District on biodiverse wildflower meadows [45,46]. Fears included water and soil acidification, an increased fire risk, disrupted water flows, and low species diversity; but, most prominent were the loss of habitat, the introduction of pests, and aesthetic damage. These concerns were shared by many participants from both groups, which indicated a collective desire for woodland expansion to be approached carefully.

As with many upland National Parks, Dartmoor abounds with protected habitats, including Rhôs pasture, blanket bog, and heathland, and much wildlife thrives in the

essentially open habitats that currently characterise the landscape. The loss of such habitats concerned the majority of key informants and one farmer. Concerns included habitat displacement, trees forming wildlife barriers, and a loss of biodiversity. However, crucially, the intent of these discussions seemed to imply not that Dartmoor should be protected from increased tree cover, but to highlight the level of care that is required when selecting sites, species, and sourcing. There was discussion about the need to consider the importance of some of these habitats, whether it is realistic to conserve some in the changing climate, and the current lack of scrub and woodland habitats. These views echo the Royal Society for the Protection of Birds (RSPB) and the Natural Capital Committee (NCC), who argue that carbon targets must be balanced with biodiversity aims [11,24].

Whittet et al. and the CCC highlighted the ability of UK nurseries to provide enough saplings to meet national tree planting targets [6,47]. With importation considered likely, several participants mentioned the risk of introducing pests and invasive species (3KI, 1FA). The estate manager for a large landowner aims to trial growing native Dartmoor seed for establishment amid gorse. Such initiatives or a greater emphasis on natural regeneration (see Section 3.2.2) will be key in mitigating potential ecological harm.

The aesthetics of increased tree cover was discussed equally by half of the interviewees, particularly in the context of conifer monoculture.

*"I want to see more trees, but I don't want the Forestry Commission planting bloody great blocks of the stuff as they've done at Fernworthy, Soussons and other places". (KI)*

*"I'm not into Christmas trees!" (FA)*

While many could see a role for softwood plantations, it was clear they feared that, in excess, these would negatively impact landscape character. Dartmoor's Landscape Character Assessment was mentioned by two key informants [48], with one seeing it as a potential barrier and the other suggesting that it needs reviewing. In 1997, English Nature found that the value that was placed on open moorland was a significant barrier to woodland expansion on Dartmoor [20]. However, Bonn et al.'s suggestion that current landscape preferences may need to be challenged [26] was reflected in the arguments regarding aesthetic values from one key informant and one farmer:

*"The aesthetic change that [allowing Willow and Alder to grow up riparian corridors] would create should be completely disregarded; I think that is our hang up". (FA)*

*"It demands a change in our aesthetic response to landscape". (KI)*

Aesthetic preferences are not static, and they should not be regarded as an unmovable barrier [27]. Rather, they should be considered and balanced with other aims. Three key informants were keen to stress that landscapes are dynamic—something that does not align comfortably with the national park's statutory role of preserving a very specific landscape. Trees take time to grow, but, if they are planted to achieve long term policy goals, Selman argues that 'landscape tastes will prove dynamic' [27] (p. 158).

### 3.1.3. Theme 3: Willing, but Constrained

All but one interviewee wished to see some level of tree cover expansion; however, this willingness is constrained by a range of structural, cultural, and practical factors.

### Common Land

The common land tenure system was the most prevalent constraint mentioned (6KI, 5FA). Common land in England dates back to the medieval feudal system; today, common land may be owned by a local council or privately, but 'commoners' may be given certain rights to the land, such as to graze livestock. Thus, commoners on Dartmoor, whose rights pertain to grazing livestock, may regard trees as reducing capacity. Land ownership and common land tenure is paid little attention in woodland expansion literature, although the CCC's Land Use report acknowledges the difficulty of tenants to make or benefit from long-term land use changes [6]. An Environmental Land Management (ELM) trial has

been instigated by the Dartmoor Hill Farm Project, addressing the question: 'How do we develop a land management plan which works for common land?' [49]. Answers to this are urgently needed in order to avoid the gridlock situation described by many participants:

> "You're going to have to ask the commoners to give up grazing, which I know from experience isn't going to happen". (FA)

> "[Farmers are] not going to want to be planting where they've got common rights". (FA)

> "It's just so difficult and this is why there's this stasis, and why Dartmoor is like it is, I think, because nobody can take it on". (FA)

> "When you've got commoners involved that means it's got to be agreement by committee, which is very, very difficult to achieve". (KI)

A number of participants suggested that attitudes might change if the ELM scheme could reward commoners for 'public goods', such as natural regeneration and ecosystem services. However, restriction of enclosure is an added complication, which makes it difficult to encourage natural regeneration grazing exclusion. One key informant explained how it took three years to obtain permission to enclose an acre of common for peatland restoration. The Dartmoor Commons Act needs alteration if anything other than grazing is to be encouraged on common land [33].

Quantifying the impact of increased tree cover on each commoner's business will be informative, as there will be asymmetries: "In some cases, a loss of a significant amount of grazing would mean that [commoners] couldn't have as much stock and that would have an impact on their economies of scale, and that could impact on their viability; but in some cases, it would be negligible". (KI). It is perhaps simplistic to regard trees solely as a negative for commoners: they might take up space that is not used for grazing while also providing shelter and shade for livestock. Common land presents a legislative and governance challenge; thus, the work of the Dartmoor ELM trial will usefully inform visions as to how the common might work for both people and nature.

Tenancy

In some ways, the constraints of tenant farming are particularly pronounced on Dartmoor, as 67,500 acres (28% of Dartmoor) are owned by the Duchy of Cornwall. The majority of these tenancies are lifelong Agricultural Holdings Act agreements that are relatively secure in contrast to the Farm Business Tenancies made after 1995, which do not offer succession provision or as much flexibility regarding land use. However, not all farmers have such agreements, with one describing the challenges of juggling thirteen landlords. Furthermore, tenancies typically reserve the tree crop to the landlord, and the insecurity of some tenancies removes the appeal of long-term land use change [3]. Farmers' views on the benefit of trees are dependent on landownership structures: "If you're farming under your own steam, you'd look at it a little bit differently". (FA)

Change is possible, as one Duchy tenant was permitted to grow trees on his farm, with these being recognised as the tenant's crop, but he was keen to stress, "This was the only such agreement I've ever heard of". (FA). For tenants to be proactive in forging such agreements, only those that are intrinsically motivated to grow trees will do so. Current tenancy practice may constrain new agricultural policy: "We can create the best ELM scheme known to man . . . , but until there's a tenancy reform in this country, it won't work". (FA). Presently, reform in England seems to be a distant prospect, so policy-makers will need to find creative solutions that work within the present system.

Farming Policy and Culture

Farming policy and culture were referred to by several key informants and farmers. Many discussed how policy has formed the current farming culture, prioritising food production and regarding trees as an obstacle:

*"Trees were a hugely important part of the farm landscape, whereas now they are seen as an intrusion". (KI)*

*"I find it all the time with landowners and farmers—farmers, especially—because . . . they consider trees as a sort of waste of money and just locking up land". (KI)*

The farmer against woodland expansion did not want to see trees in river valleys, arguing that this would take up grazing space. This view is symptomatic of a policy, which, according to two key informants, has squeezed trees out of the landscape by framing trees as forestry, not wood pasture.

Agriculture lobbies have previously challenged woodland expansion in upland environments to maintain agricultural subsidies [20]. Here, several farmers grow trees on their land, despite not receiving subsidies, and the general narrative was that farmers want to see a change to policies that disincentivise trees. However, any policy change must be undertaken sensitively if it is not to provoke opposition:

*"In saying to them [farmers] that you're doing something wrong is a massive, massive insult . . . And this is why farmers are frustrated, because they're doing exactly what they've been told to do, but now they're being told to do something different again". (FA)*

*"We have been asked to be really good at one thing, and that's the way policy has gone". (FA)*

Any new agricultural subsidy system will need to think holistically, considering all of the ecosystem services (public goods) that are required by society. Focusing on single issues can create problems elsewhere: *"People, they gotta think of things as a whole, and we're not doing it. We're getting fixated by one way, and we're running up dead ends all the time"* (FA).

Landscape Objectives

Most of the key informants and three farmers saw competing landscape objectives as a constraint. 'Shifting baselines' and cultural images of landscape were also challenged (1KI, 1FA), citing a fashion for 'tidiness' and arguing that 'scruffiness' is better for nature.

Dartmoor's Landscape Character Assessment promotes 'open, windswept upland moors' and traditional farming practices, mentioning trees only in the river valleys [48] (p. 14). This was considered to need review (1KI). Though not directly addressing the assessment, others discussed the conflict or 'perceived' conflict between an open or wooded landscape. The framing of this conflict is essentially one of perception: the increased tree cover of all presented scenarios would still leave space for much of the open Dartmoor moorland. In studies such as this, the actual visual impact of increased tree cover may need to be communicated with images in addition to maps. The contrasting landscape objectives of different groups were also mentioned (3KI, 1FA): the public want what appears to be a 'wild' landscape, and farmers see Dartmoor as pastoral. The cultural landscape of the national park—which the DNPA is obligated to maintain under the Environment Act [23]—was often framed as a constraint for not only growing trees, but also for allowing natural processes to operate; whereas, for one key informant, it is something to be preserved: *"I think we need to recognise that pastoral character as being core to the national park purposes"*; another regards trees as sympathetic to a pastoral landscape: *"This isn't just about planting woodland. This is also about looking at trees within what we consider to be an agricultural landscape, pastoral landscape"*.

Woodland expansion has the potential to create trade-offs between different ecosystem services and competing landscape objectives; further research for exploring these is needed [22,25]. While much discussion in interviews presented trees as conflicting with landscape objectives, heritage, and culture, only two participants saw them as compatible. There is a sense that increasing tree cover will change the landscape character, when, if done sensitively, it might be complementary.

Time and Money

The costs of establishing, growing, and maintaining trees were discussed by farmers, but only four key informants. The financial concerns included: the cost of fencing and weed control, paperwork, lack of time, land values, uncertainty about funding and markets, a convoluted planning process, and the cost of employing advisers. One farmer said that other farmers would see costs as a constraint, but described how she believes that trees are worth incurring these. The discrepancy between key informants and farmers in cost discussions is noteworthy. Woodland expansion can be debated *ad nauseum*, but, if land managers cannot afford trees on their land, it will not happen: *"If we cannot afford to be green, we cannot afford to be green, can we?"* (FA). Likewise, another farmer mentioned that, despite wishing to add trees to their farm, they had paused plans after discovering that the Woodland Trust grant would cover tree costs, but not fencing to exclude grazers. How to meet the up-front cost of woodland expansion needs to be given greater attention and made practical for land managers. The potential negative impacts of woodland expansion, such as reduced land value, need further study and an evidence base [28].

*3.2. Section 2: Stakeholder Recommendations*

The stakeholders interviewed had various suggestions regarding how best to proceed. These include:

3.2.1. Species and Location

The phrase *'right tree in the right place'* arose repeatedly (6KI, 2FA), reinforcing the importance of species and location choice, a topic that many have highlighted [6,11,13,24,50,51]. Both of the interviewee groups expressed an overall preference for broadleaves, but most could see a role for conifers, either as a nurse crop or for timber. Four expressed preference for broadleaves only (2KI, 2FA). In contrast, one farmer, who runs a farm-based sawmill, believes that trees must be timber-productive and that asking farmers to grow trees for trees' sake will not win their respect. One key informant noted that the phrase *'right tree in the right place'* involves a degree of subjectivity, while another added *'for the right reason'* to the phrase. Therefor, the DNPA should clarify its objectives, so that the 'rightness' of species and location can be accurately assessed.

When presented with the 'river catchment' and 'block' scenarios, participants unanimously supported a mixed approach. Some added that trees should also be grown in hedgerows and scattered throughout the landscape and, thus, emphasised the need to *"integrate trees into the whole process"* (KI) to avoid intensification in the lower valleys. The location also included consideration of the common and some advised that tree cover should not be increased there (1KI, 2FA), and another that non-common land should be prioritised. The importance of peat for natural processes and carbon storage was raised (4KI), along with the concern that the focus on trees could cause peat to be overlooked or even damaged. This is a known concern [25] and, as much of Dartmoor's common is peat, planting there is both politically and ecologically sensitive; any woodland expansion in this area should be scrupulously assessed.

3.2.2. Natural Regeneration

Natural regeneration of woodland was a popular topic raised (7KI, 1FA), and many favoured a natural-looking, scattered approach to growing trees. Natural regeneration could be part of the answer to the complexity of the common and it is already occurring on some parts of the moor (2KI). However, again, this should happen in a way that is beneficial to commoners as well as other moor users. A system which rewards commoners for natural regeneration was proposed by two key informants.

### 3.2.3. Woodland Management

The importance of how woodlands are managed was another popular topic (5KI, 2FA) and echoes a wider concern that much woodland management for keeping trees in good health, particularly important carbon sequestration, is lacking [6,14,16].

> *"I would prioritise bringing [existing woodlands] into management and enhancing them above planting a lot of new native woodland". (KI)*

A forestry skill-shortage was raised (2KI) with forestry training increasing in value as the UK increases its tree cover. The DNPA's Management Plan contains a 'Next Generation Vision' [52], which includes developing a scheme to provide apprenticeships and internships for young people. The inclusion of forestry skills in such a scheme would provide economic opportunities for young people while improving local tree health.

### 3.2.4. Funding

All of the participants agreed that there should be sufficient government funding to support woodland expansion. Unlike in Scotland and Wales, where funding has increased, grant money paid in England almost halved between 2008–2009 (£41.7 million) and 2016–2017 (£22.5 million) [53]. Three participants argued that, because it will take decades for farmers to harvest or enjoy the other benefits of their tree crop, subsidies should be provided. Other proposals for funding included: guaranteeing funding for a set time, rewarding existing woodlands, and providing funding for maintenance and for small-scale woodland projects. One farmer described how a well-designed scheme would create rapid progress: *"If the funding and all the tax breaks and everything else was in place, [farmers] would do it PDQ [=pretty damn quick]!"* and added *"A rat, a fox and a farmer are the three most adaptable things in the world".*

Additional funding from carbon credit markets or carbon-offsetting were mainly touched-on by key informants. Those who discussed such funds were concerned that they would justify un-environmental behaviour elsewhere (2FA) and one added that reverse auctions that incentivise a *"race to the bottom"* should be avoided, and that carbon credit schemes should be quality-based (1KI). Two key informants additionally suggested that water companies should contribute *a-priori* funding, as trees confer improved water processes and, therefore, can reduce water treatment costs.

It was clear that the right financial support would enable an increase in trees. The sources and distribution of this requires careful thought and the process of accessing such funds should be made straightforward for short-of-time land managers.

### 3.2.5. Support

The need for advice and help for land managers to grow trees successfully was a frequent topic; advice that was offered by the Dartmoor Hill Project was highly valued (6KI, 4FA). Some emphasised that this support should come from advisors who have an integrated understanding of both forestry and farming businesses (4KI, 1FA). Other participants felt contrastingly that farmers already have enough understanding, and it is they who should decide whether increased tree cover will benefit their farm:

> *"I think the problem is the assumption that we don't understand trees . . . it's not about the skills . . . farmers will be in the forefront of all of that . . . They know it probably far better than some of the foresters". (FA)*

Two farmers said that they would benefit from external advice, although others felt that this was not required, and some of those might have sufficient skill to be right. While some might not need advice, an inspection process for verifying appropriate delivery against funding should be devised.

### 3.2.6. Governance

Opinions regarding who should be responsible for woodland expansion planning were varied. Some (5KI, 1FA) supported the idea of a partnership of all interested parties,

others (2KI, 3FreA) thought that an organisation, such as the DNPA or FC, should oversee matters, or believed (2FA) that land managers would simply take action if incentives were available. The FC's role is largely managerial and regulatory, and it is not currently responsible for meeting tree planting targets. While national park management plans can create a vision, they lack legal power—something which the Landscape Review has called to change [41]. Thus, a partnership which empowers land managers and local people seems the best way forward.

Whichever scheme emerges, there was strong support from both groups for a simpler planning process. Several called for a landscape scale plan mapping the area's aims; with specific recommendation that the DNPA Management Plan make these aims clear (2KI). It was broadly hoped that, with a transparent decision-making framework, land managers would be able to easily assess how they fit into the park's aims and more quickly receive approval.

## 4. Conclusions

The commitment to a significant expansion of woodland areas over coming decades brings into play the interests of many stakeholders; understanding those most directly involved in its execution is key to the success of any tree planning programme. The stakeholder mapping and engagement process used here revealed enthusiasm for woodland expansion as a policy project, although this was accompanied by concerns that it must be done carefully and with respect for a range of constraints, mainly financial and ecological, as well as the cultural nuance of landscape setting. A thorough process of engagement should consult widely and, in addition to the active participants that are involved here, should seek to include the tourism sector and local residents for a fuller view of the social and cultural impacts of such woodland expansion.

While Dartmoor cannot represent all national parks, there are common issues both within the UK and in other countries. These include: a duty to protect cultural heritage that may conflict with environmental aims; pre-existing management plans that provide specified landscape visions; sensitive habitats; and, complicated legislation concerning land tenure and grazing rights. With woodland expansion likely to be a key climate mitigation mechanism internationally, those who manage other protected landscapes can learn from the UK experience being reported here. Increasing tree cover in a national park is, in practice, complicated, but with the voices of stakeholders heard, the appetite for planting more trees suggests that identifying the 'right trees, right places' would be positive for their many users. Towards this effort are the following key messages:

From farmers:

- Woodland expansion must be financially viable for land managers. Money must be available for maintenance as well as tree planting.
- From key informants:
- A partnership of organisations should work together to achieve landscape aims. The Dartmoor National Park Authority could take a faciliatory role and, ideally, would have statutory powers to enable delivery.
- A publicly agreed map indicating areas for woodland expansion should be published. Highlighted areas should receive a streamlined application process.
- The Landscape Character Assessment should be re-evaluated in the context of the climate and ecological crises.
- From both groups:
- New agricultural policies need to adequately reward the ecosystem services that are conferred by trees.
- Planning processes are convoluted and off-putting so must be streamlined for land managers.
- Integrated advice, such as that already offered by the Dartmoor Hill Farm Project, is extremely helpful; more should be available.

- To meet the demand for forestry skills, the forestry sector should create apprenticeships and internships for young people or those who may be looking for new opportunities amid the Covid-19 pandemic.
- The Dartmoor National Park Authority and forestry sector should consult local residents and the tourism sector to obtain additional perspectives regarding how the benefits of increased tree cover can be maximised.

**Author Contributions:** Conceptualization, O.F. and C.P.; methodology, O.F.; software, O.F.; validation, O.F.; formal analysis, O.F.; investigation, O.F.; resources, O.F.; data curation, O.F.; writing—original draft preparation, O.F.; writing—review and editing, O.F.; C.M.C. and C.P.; visualization, O.F. and C.M.C.; supervision, C.P. and C.M.C.; project administration, O.F.; funding acquisition, n/a. All authors have read and agreed to the published version of the manuscript.

**Funding:** This research received no external funding.

**Data Availability Statement:** The data presented in this study were collected under the auspices of the Science, Engineering and Technology Research Ethics Committee (SETREC) of Imperial College London. Any request for access to the anonymised data can be made to the corresponding author.

**Acknowledgments:** The authors would like to thank all interview participants for generously giving their time.

**Conflicts of Interest:** The authors declare no conflict of interest.

## Abbreviations

| | |
|---|---|
| CCC | Climate Change Committee (UK) |
| CAT | Centre for Alternative Technology |
| Defra | Department for Environmental, Food and Rural Affairs (UK) |
| DNPA | Dartmoor National Park Authority |
| ELMs | Environmental Land Management scheme (Great Britain) |
| FC | Forestry Commission (England) |
| FOE | Friends of the Earth |
| GHG | Greenhouse gas |
| NCC | Natural Capital Committee (UK) |
| NFU | National Farmers' Union (UK) |
| RSPB | Royal Society for the Protection of Birds |

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
