# Peer review of "Woodland Expansion in Upland National Parks: An Analysis of Stakeholder Views and Understanding in the Dartmoor National Park, UK"

_land, doi:10.3390/land10030270_

Round 1
Reviewer 1 Report
This is a well-written and timely piece of research about woodland expansion to mitigate climate change/deliver climate change targets. Your case study is particularly interesting as the potential conflicts between agriculture and woodland creation can be examined in detail. You provide a detailed policy and academic literature review at the outset and use an engaging map-based method to present scenarios to participants. The results are presented clearly and the stakeholder analysis explains the participant selection. Direct quotes are integrated effectively into the results narrative and this helps to understand the different perspectives/priorities of the participants. Figure 4 provides a very helpful flow chart of the cautions shared by participants and this is helpful for an international readership.
A minor observation is that the focus is on UK CC targets and the case study is located in England. At times, you could acknowledge different approaches in other parts of the UK, particularly Scotland, where there is a different context in terms of landscape-scale planning for woodland creation (e.g. planned Regional Land Use Partnership and Land Use Frameworks that will formalise landscape-scale planning and the recently announced Scottish Government pilots will include the two national parks). The land tenure pattern and involvement of communities through land reform legislation and policy is also distinct within the UK as a whole (for example, you talk about common land in 3.3.1 but the system in England is different to Scotland - in line 372 there has arguably been more research on this in Scotland, for example).
To address this, I would suggest acknowledging this variation in contexts towards the end of Section 1, perhaps in relation to Tokarski and Gammon's claim about needing adequate debate - arguably, this is beginning to be formalised in Scotland. Throughout the article, check for any statements that are applied to UK yet may have differences in devolved nations. You might also reflect on the extent to which your conclusions and key messages apply to the whole of the UK, or whether some of them are more relevant to England where landscape-scale partnerships to date are slightly different to the Scottish context, for example.
Some other minor points and typos:
- p.2 line 61 two full stops
- p.3 line 89 - the 'British' uplands?
- p.3 line 102 typo 'likley'
- p.4 line 133 typo 'blocks' - should be 'block'?
- p.7 Figure 3 - could the blocks in the figure be shaded/cross-hatched as well for those reading in black and white print version?
- There is some inconsistency in use of italics/non-italics for direct quotes. e.g. p.11 line 311
- p.10 line 289 - Skene's is highlighted - not sure why
- p.14 line 418 - yet reform is a reality in Scotland - again, some clarity on the English focus would be helpful.
- p.16 line 534 - it would be helpful for the international reader to have a brief understanding of what support is available for woodland expansion (in England?) - again, this is different in England and devolved nations?
Author Response
Dear Reviewer,
Thank you for your time in considering this manuscript for publication in Land. We appreciate your insight and experienced advice on the manuscript, and how it fits into the wider scope of the journal. It’s encouraging to read your comments about this being a timely piece of research. We have been able to incorporate changes to reflect most of the suggestions provided by the reviewers. These changes have been made to the manuscript, and a point-by-point response to your comments is outlined below. We hope that this has made the work into something that satisfies the requirements of the journal.
In response to your main observation about making clear the different contexts of the UK countries, we have added a note about devolution early in the paper and have added other comments about country-specific policies etc where relevant. We hope that this is clear to both the domestic and international audiences despite the complex policy landscape.
Thank you again for your generous help and advice.
Yours faithfully,
Olivia FitzGerald (corresponding author, approved by all authors)
L.93-96
In response to clarifying the UK devolved policy context, I have added the following
This debate might take different forms depending on which UK nation (England, Northern Ireland, Scotland or Wales) is involved. Forestry is a devolved issue, with each nation able to set its own policies amid its particular land tenure and landscape planning contexts.
Some other minor points and typos:
- 2 line 61 two full stops done
- 3 line 89 - the 'British' uplands? done
- 3 line 102 typo 'likley' done
- 4 line 133 typo 'blocks' - should be 'block'? done
- 7 Figure 3 - could the blocks in the figure be shaded/cross-hatched as well for those reading in black and white print version? Nice suggestion, done
- There is some inconsistency in use of italics/non-italics for direct quotes. e.g. p.11 line 311 There was a logic (italics when in paragraphs and non-italics when standalone quote), but maybe that's not obvious so I've changed them all to italics.
- 10 line 289 - Skene's is highlighted - not sure why done (it was a hangover from previous referencing software)
- 14 line 418 - yet reform is a reality in Scotland - again, some clarity on the English focus would be helpful.
Presently, reform seems a distant prospect, so policy-makers will need to find creative solutions which work within the present system. --> Presently, reform in England seems a distant prospect, so policy-makers will need to find creative solutions which work within the present system.
- 16 line 534 - it would be helpful for the international reader to have a brief understanding of what support is available for woodland expansion (in England?) - again, this is different in England and devolved nations?
Added: Unlike in Scotland and Wales where funding has increased, grant money paid in England almost halved between 2008-2009 (£24.1 million) and 2017-2018 (£13.5 million) [53].
Reviewer 2 Report
I consider that the theme dealt in this manuscript is interesting as one of case examples of sustainable land use management to meet both social and environmental needs. The manuscript is worth publishing if several points are revised and improved.
P6. Figure 2
Where are the selected key informants located in Figure 2 ? In addition, where are farmers located in the figure?
P6. 2.4 Interviews
The author should note the date the interview was conducted.
P7. L.178-183
It is not clear how the three main themes such as enthusiasm, caution and willingness are extracted.
P18 4.Conclusions
Is is not clear how the both results of section 1 and 2 were connected and integrated, and as a result came to the conclusions. Though many opinions of the key informants were indicated in the Results and Discussion, I am not sure how they were summarized and lead to the conclusions. The authors need to logically indicate the thinking process from the results to the conclusions.
Author Response
Dear Reviewer,
Thank you for your time in considering this manuscript for publication in Land. We appreciate your insight and experienced advice on the manuscript, and how it fits into the wider scope of the journal. We have been able to incorporate changes to reflect most of the suggestions provided by the reviewers. These changes have been made to the manuscript, and a point-by-point response to your comments is outlined below. We hope that this has made the work into something that satisfies the requirements of the journal.
In response to reviewers’ comments and suggestions, effort has been made to make clear the different contexts of the UK countries: we have added a note about devolution early in the paper and have added other comments about country-specific policies etc where relevant. We hope that this is clear to both the domestic and international audiences despite the complex policy landscape.
Thank you again for your generous help and advice.
Yours faithfully,
Olivia FitzGerald (corresponding author, approved by all authors)
P6. Figure 2
Where are the selected key informants located in Figure 2 ? In addition, where are farmers located in the figure?
To make this clearer I have colour-coded the participants and have written 'Interviewees are representative of the blue (farmers) and yellow (key informants) stakeholder boxes.'
P6. 2.4 Interviews
The author should note the date the interview was conducted.
Because there were several interviews, I have written (new text in bold): 'Participants were interviewed remotely (between 29 June and 22 July 2020), interviews were recorded, transcribed by Otter.ai software and then summarised for further analysis.'
P7. L.178-183
It is not clear how the three main themes such as enthusiasm, caution and willingness are extracted.
I have referred back to the methodology section by inserting the phrase: ‘Thematic analysis led to identification of.'
P18 4.Conclusions
Is is not clear how the both results of section 1 and 2 were connected and integrated, and as a result came to the conclusions. Though many opinions of the key informants were indicated in the Results and Discussion, I am not sure how they were summarized and lead to the conclusions. The authors need to logically indicate the thinking process from the results to the conclusions.
I've edited a couple of sentences (see below) to make it more explicit how we arrived at our conclusions and key messages (see below).
L.594-595
'They mostly expressed strong enthusiasm for woodland expansion as a policy project.' --> 'Thematic analysis highlighted strong enthusiasm for woodland expansion as a policy project.'
L.598-600
'This work reinforces the message that care should be taken, that all stakeholder groups should be consulted, but not that ambition should be reduced.' --> 'This work reinforces the message from the literature that care should be taken [9-11, 24], that all stakeholder groups should be consulted [26, 35], but not that ambition should be reduced [4-6].'
Reviewer 3 Report
Thanks for submitting your manuscript for consideration for publication. The sharing of the perspective of the research participants is interesting.
There are quite a few typos and small errors that you need to address. I have pointed out some of these in the attache manuscript, but you need to look throughout the paper for other similar errors.
More explanation of the interesting land tenure arrangements in place in the study sites is needed. You only mention these in passing, but readers not familiar with the arrangements will not gain much. Such explanation will also make the paper more relevant to this journal and its readers.
It would also be good to have more elucidation of the debates on the pros and cons of tree planting. It is mentioned a few times that there are some negative views of tree planting as a response to climate change and the potential impacts on water and bio diversity, but these debates are not really elaborated.
While the empirical information is interesting, there is little engagement with theoretical debates and it is written from a rather UK centric perspective and this reduces its value to an international journal. I leave it to the editors to decide on suitability.

Author Response
Dear Reviewer,
Thank you for your time in considering this manuscript for publication in Land. We appreciate your insight and experienced advice on the manuscript, and how it fits into the wider scope of the journal. We have been able to incorporate changes to reflect most of the suggestions provided by the reviewers. These changes have been made to the manuscript, and a point-by-point response to your comments is outlined below. We hope that this has made the work into something that satisfies the requirements of the journal.
In response to your suggestions for making the paper more relevant and readable for an international audience, effort has been made to make clear the different contexts of the UK countries: we have added a note about devolution early in the paper and have added other comments about country-specific policies etc where relevant. We hope that this is clear to both the domestic and international audiences despite the complex policy landscape.
Thank you again for your generous help and advice.
Yours faithfully,
Olivia FitzGerald (corresponding author, approved by all authors)
L.371-375
The common land tenure system was the most prevalent constraint mentioned (6KI, 5FA) and commoners, who have grazing rights for livestock, may regard trees as reducing capacity. -->
The common land tenure system was the most prevalent constraint mentioned (6KI, 5FA). Common land in England dates back to the medieval feudal system; today, common land may be owned by a local council or privately, but ‘commoners’ may be given certain rights to the land, such as to graze livestock. Thus commoners on Dartmoor, whose rights pertain to grazing livestock, may regard trees as reducing capacity.
It would also be good to have more elucidation of the debates on the pros and cons of tree planting. It is mentioned a few times that there are some negative views of tree planting as a response to climate change and the potential impacts on water and bio diversity, but these debates are not really elaborated.
I've added the following sentence which is intended to reference succinctly some of the negative views/dangers of tree planting:
L.55-58
Conservationists warn against focusing narrowly on carbon when using nature-based solutions, emphasising a holistic approach that avoids disrupting natural processes or damaging protected habitats [11, 24].
While the empirical information is interesting, there is little engagement with theoretical debates and it is written from a rather UK centric perspective and this reduces its value to an international journal. I leave it to the editors to decide on suitability.
Indeed, this paper is rather UK-centric and efforts have been made to explain the idiosyncrasies of the UK policy and land tenure context. Despite the UK focus, we hope that the issues explored will be relevant to international readers where upland landscapes encounter similar pressures. Furthermore, we hope that our insistence that stakeholders are consulted properly will resonate universally.
Made specific changes to comments made on the manuscript (typos etc, added UK/Great Britain etc to some of the abbreviations)
L.55 - 'arising timber' – changed to ‘timber products’
L.99-101 - re comment about it getting repetitive…I think this is drawing attention to the key issue before setting up the structure of the rest of the paper, so I've decided to keep this in – I hope that’s excusable!
L.123
This is privately owned but other locals have certain common rights such as livestock grazing and turf cutting for domestic use; there is also provision for some public access on foot and on horseback [33].
--> In accordance with common land in England and Wales, this is privately owned but other locals have certain common rights such as livestock grazing and turf cutting for domestic use; there is also provision for some public access on foot and on horseback [33].
L.177-179
How's this?
The key informants engaged more readily with the process than farmers, for whom future tree cover extent appeared to feel less immediate. -->
The key informants engaged more readily with the process than farmers, for whom the issue of increasing trees may be a less immediate concern than the daily running of an agricultural business.
L.199-200 - The language of this quote is quite colloquial which might not work well for an international journal. Perhaps an explanation of the colloquial terms could be useful? I will leave it for editors to decide.
L.291-292
Skene’s concern that there may be an over-reliance on tree-planting within climate policy was voiced by five interviewees (4FA, 1KI) [12]. --> Concern, raised in the literature, that there may be an over-reliance on tree-planting within climate policy was voiced by five interviewees (4FA, 1KI) [12].
L.297-298
Brown’s concern that a rush to expand tree cover could lead to inappropriate land use change was raised by two farmers, --> Concern, as outlined by Brown [13], that a rush to expand tree cover could lead to inappropriate land use change was raised by two farmers,
L.308-309
Response to reviewer’s question: the participant is saying that Dartmoor isn't a productive environment for food production so other land uses should receive more consideration.
L.410-414
In some ways, the constraints of tenant farming are particularly pronounced on Dartmoor as 67,500 acres (28% of Dartmoor) are owned by the Duchy of Cornwall. The majority of these tenancies are lifelong Agricultural Holdings Act agreements which are relatively secure. --> The majority of these tenancies are lifelong Agricultural Holdings Act agreements which are relatively secure in contrast to the Farm Business Tenancies made after 1995, which do not offer succession provision or as much flexibility regarding land use.
L.555-567
Two key informants additionally suggested that water companies should contribute a-priori funding as trees confer improved water processes. --> Two key informants additionally suggested that water companies should contribute a-priori funding as trees confer improved water processes and can therefore reduce water treatment costs.
CONCLUSION
Reviewer comment: I wouldn't end with these. It makes it more of a project planning document than a journal article.
We do recognise this perspective and would understand if editors decide to cut the key messages.
Round 2
Reviewer 2 Report
The manuscript is fully revised and qualified for publication.
Author Response
Dear Reviewer, Thank you very much for your attention once again. We are very pleased that you have approved our revisions. We have made the conclusion more concise; we think it now reads more clearly and succinctly. Your reviews have been much appreciated, Olivia FitzGerald